# Comparative Analysis and Phylogenetic Insights of Cas14-Homology Proteins in Bacteria and Archaea

**DOI:** 10.3390/genes14101911

**Published:** 2023-10-06

**Authors:** Numan Ullah, Naisu Yang, Zhongxia Guan, Kuilin Xiang, Yali Wang, Mohamed Diaby, Cai Chen, Bo Gao, Chengyi Song

**Affiliations:** College of Animal Science and Technology, Yangzhou University, Yangzhou 225009, China; dh18056@stu.yzu.edu.cn (N.U.); dx120180101@yzu.edu.cn (N.Y.); mx120190652@yzu.edu.cn (Z.G.); mx120220867@stu.yzu.edu.cn (K.X.); dx120180099@yzu.edu.cn (Y.W.); dh18035@yzu.edu.cn (M.D.); 007302@you.edu.cn (C.C.); bgao@yzu.edu.cn (B.G.)

**Keywords:** Cas14, Cas proteins, Cas14-homology protein, Cas14 mining, phylogenetics

## Abstract

Type-V-F Cas12f proteins, also known as Cas14, have drawn significant interest within the diverse CRISPR-Cas nucleases due to their compact size. This study involves analyzing and comparing Cas14-homology proteins in prokaryotic genomes through mining, sequence comparisons, a phylogenetic analysis, and an array/repeat analysis. In our analysis, we identified and mined a total of 93 Cas14-homology proteins that ranged in size from 344 aa to 843 aa. The majority of the Cas14-homology proteins discovered in this analysis were found within the *Firmicutes* group, which contained 37 species, representing 42% of all the Cas14-homology proteins identified. In archaea, the *DPANN* group had the highest number of species containing Cas14-homology proteins, a total of three species. The phylogenetic analysis results demonstrate the division of Cas14-homology proteins into three clades: Cas14-A, Cas14-B, and Cas14-U. Extensive similarity was observed at the C-terminal end (CTD) through a domain comparison of the three clades, suggesting a potentially shared mechanism of action due to the presence of cutting domains in that region. Additionally, a sequence similarity analysis of all the identified Cas14 sequences indicated a low level of similarity (18%) between the protein variants. The analysis of repeats/arrays in the extended nucleotide sequences of the identified Cas14-homology proteins highlighted that 44 out of the total mined proteins possessed CRISPR-associated repeats, with 20 of them being specific to Cas14. Our study contributes to the increased understanding of Cas14 proteins across prokaryotic genomes. These homologous proteins have the potential for future applications in the mining and engineering of Cas14 proteins.

## 1. Introduction

The Clustered Regularly Interspaced Short Palindromic Repeats (CRISPR) and CRISPR-associated Proteins (Cas) are a part of the bacterial and archaeal defense system. These proteins have the capability to eliminate invading viruses and mobile genetic elements (MGEs) and have also been used in gene editing [1,2]. In the past decade, researchers have primarily relied on Cas proteins that have over 1000 amino acids, such as Cas9 and Cas12a, for gene-editing purposes [3,4,5]. However, these longer proteins have encountered various challenges, such as limitations in delivery and low efficiency [6]. To address these challenges, miniature Cas proteins, such as CasΦ, Cas13, and Cas14, have been developed for gene editing and regulatory purposes. These proteins have a size between approximately 400 and 700 amino acids and consist of the RuvC nuclease domain [7,8,9,10,11]. The development of these smaller proteins holds significant importance for gene therapy applications as they can be encapsulated within FDA-approved adenoviral vectors (AVV) thanks to their smaller size [12].

Cas14 (Cas12f), a member of the class-2 type-V-F CRISPR-associated effector nuclease family, has been discovered through an analysis of genomic and metagenomic databases [13]. By conducting a comprehensive analysis of extensive terabase-scale sequencing datasets, the group uncovered a putative group of single-effector Cas proteins. This process involved meticulously sifting through the datasets to identify genes with a RuvC-like domain located proximal to CRISPR loci. The resulting collection of identified proteins signifies a significant advancement in the field due to their highly novel nature. Phylogenetically, Cas14 shares similarities with the bacterial RuvC-containing proteins C2c10 and C2c9, which are typically found near Cas arrays rather than Cas genes [13]. Initially discovered in uncultured archaea, these miniature effectors are capable of making RNA-guided single-strand (ssDNA) breaks in the target DNA without the need for any PAM sequence [13]. However, later studies have demonstrated that these effectors can induce double-stranded breaks (DSB) using the T-rich PAM sequence [8,14,15]. The majority of identified Cas14 proteins were ineffective in shielding Escherichia coli from invading double-stranded DNA. Out of the ten Cas14 proteins tested, only AsCas12f and SpCas12f exhibited the ability to mediate plasmid interference in *E. coli* [14,15,16]. Several studies have recently focused on gRNA engineering approaches in order to utilize Cas14 as a tool for editing genes in mammalian cells. Although several Cas14 proteins were found to cleave double-stranded target DNA in bacteria, no detectable activity in mammalian cells has been reported. Through the optimization of guide gRNAs and multiple rounds of protein engineering and screening, Cas12f variants were created and given the name CasMINI [17]. These variants demonstrated improved activity in the mammalian genome. Furthermore, not only have the biochemical features of SpCas12f been deciphered, but its effectiveness as a gene-editing tool in plants and human cells has also been demonstrated [9]. Similarly, subsequent gRNA engineering in SpCas12f has improved its efficiency in human cells. With the engineered gRNA, SpCas12f was able to achieve editing efficiencies comparable to that of Cas12a, thus expanding the CasMINI toolbox [17,18]. 

The Cas14 protein structure was determined through X-ray crystallography [18,19]. The study revealed that the Cas12f protein is a homodimer with two identical subunits comprising a nuclease domain and a helicase-nuclease domain. The nuclease domain of Cas12f encompasses the active site for DNA cleavage, consisting of two metal ions: a magnesium ion and a manganese ion. The helicase–nuclease domain is responsible for unwinding the target DNA and stabilizing the protein-DNA complex.

Bioinformatics tools have played a crucial role in the discovery and development of CRISPR-Cas as a gene-editing technology [20]. Recent studies have focused on discovering and characterizing novel miniature type-V Cas12f nucleases that exhibit diverse protospacer adjacent motif (PAM) preferences [21]. A comparative phylogenetic analysis of Cas proteins is important for understanding the diversity and evolution of CRISPR-Cas systems, as well as discovering novel Cas proteins that have potential applications in gene editing and other biotechnological fields [22,23]. Bioinformatics tools, such as BLAST, MUSCLE alignment algorithm, and neighbor-joining consensus trees based on the Jukes–Cantor model, can be utilized for performing comparative phylogenetic analyses of Cas proteins [24,25]. These analyses aid in the classification of CRISPR-Cas systems into distinct classes, types, and subtypes based on the conservation of Cas protein sequences and the architectural features of Cas loci. Additionally, a comparative phylogenetic analysis aids in the identification of novel Cas proteins by evaluating the preservation of Cas protein sequences. Furthermore, it can be employed to reconstruct the ancestral gene content and track the gain and loss of genes during the evolution of Cas proteins [26,27]. With the extensive diversity of CRISPR-Cas genes in the prokaryotic genome and the growing number of genomic and metagenomic sequencing data of bacteria and archaea being submitted to public databases, manually identifying, classifying, and tracking their evolutionary background is impractical [27,28]. Therefore, various automated approaches have been developed to enhance the gene-editing toolbox of Cas proteins [29,30,31,32,33,34]. These bioinformatics programs can predict the presence and location of CRISPR-Cas genes within a genome, as well as identify their specific class and subtype, shedding light on their evolutionary origins and potential functional roles. 

Previous phylogenetic analyses of the Cas14 protein family have yielded the identification of multiple Cas14 variants. For instance, recent studies have led to the discovery of three novel Cas12f effectors, referred to as μCas, which were derived from metagenome-assembled genomes (MAGs) found in ruminant microbiomes. In their study, Kong et al. characterized six CRISPR-Cas12f1 systems and specifically chose OsCas12f1 and RhCas12f1 for further investigation. [35]. OsCas12f1 recognizes a 5′ T-rich protospacer adjacent motif (PAM), while RhCas12f1 recognizes a 5′ C-rich PAM. To enhance their editing efficiency and broaden the recognition range of PAMs, protein and sgRNA engineering techniques were employed to develop advanced variants of OsCas12f1 (enOsCas12f1) and RhCas12f1 (enRhCas12f1). These enhanced variants showed increased editing efficiency and wider PAM recognition compared to the engineered variant Un1Cas12f1 (Un1Cas12f1_ge4.1) [35]. Additionally, an inducible-enOsCas12f1 construct was developed by fusing the destabilized domain with enOsCas12f1. Through the delivery of a single adeno-associated virus, the in vivo activity of inducible-enOsCas12f1 was demonstrated. These findings highlight the ongoing exploration of the diverse repertoire of Cas proteins and their potential functions within microbial communities. [21] However, these studies have not identified any discernible distinctions among the various subtypes of Cas14. In this study, our goal was to elucidate the diversity of Cas14-homology proteins and to characterize the domain disparities within the Cas14 protein family. By thoroughly analyzing publicly available databases, we successfully retrieved a set of Cas14-homology proteins. Our findings demonstrated the domain organization, sequence similarity, species distribution, and repeat analysis of these proteins, offering novel insights into the intricate and heterogeneous structure of the Cas14 family. Moreover, mining Cas14-homology proteins holds the potential to serve as a valuable reference for further functional enhancement in Cas14 engineering.

## 2. Materials and Methods

### 2.1. The Cas14 Mining 

All available viral and prokaryotic genomes were downloaded from the NCBI database. To generate protein sequences, the transeq program of Emboss software was utilized. This program is capable of translating in any of the three forward or three reverse sense frames or in all six frames. The Cas14 sequences were obtained from reliable references. The CTD of Cas14, which includes motifs of RuvC segment I, II, and III, as well as recognition lobe 2 (REC2), Lid, and Nuc (the target nucleic acid binding), were aligned to create a hidden Markov model (HMM) profile named as Cas14-CTD.hmm. This profile is provided in the supplementary file. The Cas14 sequences were initially searched using the hmmsearch program of HMMER3 [36] in the translated proteins with Cas14-CTD.hmm with an evalue of 1^e-10^. Then, the protein sequences with 300 amino acids upstream and 100 amino acids downstream flanks of the hmmsearch hits were extracted. These sequences were then filtered using the local BLASTP program against all available Cas14 sequences, selecting candidates with an e-value higher than 1^e-30^. Subsequently, the genomic sequences with 20 kb flanks of these candidates were extracted and submitted for the annotation of the Cas14 protein and CRISPR array using the CRISPRCasTyper program [37]. A size filter was applied to the obtained Cas14 protein, only retaining proteins with a length greater than 300 amino acids. The presence of the CTD was confirmed in the sequences using alignment with MaFFT. Finally, the putative Cas14 proteins (>300 amino acids) were subjected to further domain and phylogenetic analyses.

### 2.2. Domain and Phylogenetic Analysis

The obtained Cas14 subfamilies, together with the reference sequences of Cas14, were submitted for alignment using the E-INS-I method from the MAFFT software [38]. The phylogenetic tree was inferred by using the maximum likelihood method in the IQ-TREE program [39], with a best-suited aa substitution model selected by ModelFinder, and the ultrafast bootstrap approach with 1000 replicates was applied.

### 2.3. CRISPR Array/Repeats Prediction 

The sequences identified through the phylogenetic analysis as Cas14 were subjected to a CRISPR Array analysis. Briefly, the prediction and classification of Cas operons and their spacers were carried out by CRISPRCasTyper 1.2.4 (https://github.com/Russel88/CRISPRCasTyper, accessed on 5 August 2023) [37]. Briefly, the extended 20 kb (per side) nucleotide sequences were taken for the proteins identified as Cas14 through blastp, and subsequent phylogenetic analyses were carried out with the CRISPRCasTyper.

## 3. Results

### 3.1. Classification of Cas14-Homology Proteins

All the mined Cas14-homology proteins were submitted for the primary phylogenetic analysis, and the minor branches in the tree with less than four sequences were removed, which was implemented to exclude rare sequences. Then, all the sequences (93 sequences, Appendix A) that passed the threshold were used for the IQ-Tree reconstruction. The ultimate phylogenetic tree classified the mined Cas14 proteins, along with the Cas14 proteins from published data, into three primary branches (Cas14-A, Cas14-B, and Cas14U) with strong support, as evidenced by bootstrap values exceeding 70%. The branch Cas14U (26 sequences) seems to be the most prominent, while the remaining branches, Cas14A (58 sequences) and Cas14B (9 sequences), exhibit potential for further subdivision into smaller branches. Within Cas14A, there are two sub-branches named Cas14A-I and Cas14A-II. Analogously, Cas14B is segregated into two subbranches, Cas14B1 and Cas14B2 (see Figure 1). In the analysis, the Cas1 protein, which is a universally conserved component of the CRISPR prokaryotic immune defense system, served as an outgroup for tree rooting [40] (Figure 1).

### 3.2. Domain Organization of Cas14-Homology Proteins

The majority of the Cas14-homology proteins we identified belong to the typical miniature Cas protein category, with lengths ranging from 344 aa to 843 aa. Varied sizes were observed among the Cas14-A, Cas14-B, and Cas14U clades, with Cas14-B having the longest average length (~590 aa) and Cas14-A being the shortest, with an average size of ~447 aa (Appendix A). The crystal structure of Cas14, which was recently determined [19], provides insight into its recognition and cleavage mechanism [41]. By aligning the mined Cas14 proteins with the reference Cas14 proteins, Un1Cas12f1, the domain organization of the three clades of Cas14 (Cas14-A, Cas14-B, and Cas14U) was inferred (Appendix A). This analysis reveals that the NTD in all three clades contains a Rec domain sandwiched between two WED domains (Figure 2A). At the CTD of Cas14, three RuvC domains are found in conjunction with the Rec2, lid, and Nuc domains. Compared to the NTD, the CTD of all three clades exhibits a high degree of conservation. The conservation of the CTD in all three branches may be attributed to the presence of RuvC (I, II, III) segments, implying a potential shared excision mechanism among these proteins. Based on the alignment, some differences in the domain organization can be observed among the three clads. Notably, Cas14-B seems to have a different domain organization, especially at the NTD. Moreover, domain differences within the clades were also observed; for instance, Cas14-A1 and Cas14-II seem to have differences at the WED and REC domains at the NTD (Appendix A).

The overall sequence identity of the Cas14-homology proteins analyzed in this study was generally low, ranging from 6% to 70%, with an average identity of 18% (Figure 3A). Furthermore, a more detailed analysis of the sequence identity within each clade revealed that the Cas14-U clade had a sequence identity range of 7% to 70% and an average identity of 27% (Appendix A). In terms of the specific clades, the Cas14A clade exhibited an average sequence identity of 26%, ranging from 13% to 99%, while the Cas14-B clade displayed an average sequence identity of 36%, ranging from 10% to 97% (Appendix A).

### 3.3. Putatively Functional Cas14 Proteins

The genomic coordinates flanking the investigated Cas14-homology proteins were screened using the standalone version of CRISPR-Castyper to identify the presence of CRISPR arrays/repeats. A 20 kb nucleotide sequence on each side of the target region was subjected to analysis. The results revealed the existence of 44 sequences with arrays (as presented in Appendix A), with 20 of these sequences being classified as type-V-F CRISPR-Cas (refer to Table 1). Notably, the data showcased a diverse range in the size of the repeat-spacer arrays, varying from 3 to 19 spacers (as summarized in Appendix A). The CRISPR systems exhibited an average of nine repeats per locus. Additionally, the length of the repeat sequences was analyzed and displayed an average of 30 bp, ranging from 23 bp to 38 bp (as exemplified in Figure 3). Furthermore, an alignment of the putative Cas14 protein indicated that these putative proteins share major domains, particularly the RuvC domain that functions as an endonuclease domain that is present in the Cas14 reference protein UniCas12f1 (refer to Figure 4). Although this observation suggests that our identified homology proteins are putatively functional, it is important to note that Cas proteins require substrate recognition, inhibition, and excision, which involve multiple domains and neighboring arrays. Therefore, a thorough comparative functional comparison may necessitate further analysis. In contrast to the other Cas14-homology proteins identified in our study, the putative Cas14-homology proteins possess conserved domains both at the CTD and NTD. This suggests that these proteins may potentially be functionally active and further supports the presence of Cas14-associated arrays in our investigation.

### 3.4. Distribution of Ca14-Homology Proteins in Bacteria and Archaea

Our analysis revealed that mined Cas14 is present in both bacterial and archaeal species (Table 2). The *Firmicutes group* had the highest number of mined Cas14 homology proteins, with 37 species accounting for 42% of the total mined Cas14-homology proteins. The *Actinobacteria group* accounted for the second-highest number of species with mined Cas14, representing 18% of the total. Among the archaeal groups, there were a total of 10 species found to have Cas14 homology proteins. The *DPANN group* had the highest number of species with mined Cas14, representing 90% of the archaeal species having the protein. It accounted for 10% of all species in our analysis with mined Cas14-homology proteins (Table 2).

### 3.5. Distribution of Putatively Functional Cas14 Proteins

Our analysis uncovered a prominent distribution of putative Cas14 among various species. The results demonstrate that the *Firimutes* clade comprises the majority of species with putative Cas14 genes, representing 16% of all species in this category. Notably, within the archaeal clades, the *DPANN* group exhibits the highest number of species that harbor putative Cas14, accounting for 12% of all species with this gene (Table 2).

## 4. Discussion

Among the diverse CRISPR-Cas systems, class 2 is the most versatile due to its utilization of a single-effector protein. Particularly, miniature Cas proteins have gained significant importance in the field of gene therapy due to their smaller size in contrast to the commonly utilized Cas proteins such as Cas9 and Cas12a. The recent achievement in mining and gRNA engineering of miniature Cas12, specifically Cas14 (Cas12f), which is smaller than any other Cas proteins discovered thus far, has sparked a growing interest in discovering and characterizing these compact Cas proteins. Consequently, numerous studies have been conducted to enhance our comprehension of their structure, cleavage, and binding and compare the structure of Cas14 with its closely related counterpart in the Cas12 family [19,41]. For example, a detailed comparison has been conducted between Cas14 and CasΦ to compare their characteristics [42]. The comparison indicates that Cas14 and Cas12j, due to their smaller size, are less efficient than larger Cas endonucleases in cleaving target DNA. Their smaller size leads to slower cleavage kinetics and reduces genome-editing outcomes in human cells, as evidenced by multiple studies [14,17,43] and confirmed experimentally in a recent study [44]. This may be due to the absence of stabilizing contacts between the smaller proteins and the RNA –DNA heteroduplex. However, it has been demonstrated that modifying the Cas12f structure to enhance these contacts increases its efficiency in gene activation and editing [17,33]. Furthermore, the slower kinetics of the miniature Cas12 proteins may also impact their specificity, given that these proteins are designed to exhibit a certain degree of non-specificity towards their target DNA. However, preliminary reports indicate that they may demonstrate a limited tolerance for mismatches in the PAM-proximal region of the target [35].

The crystal structure of Cas14, which elucidates its recognition and cleavage mechanism, was recently determined [18,30]. According to the first study, the Cas12f protein is a homodimer with two identical subunits, each containing a nuclease domain and a helicase–nuclease domain [18]. The NTD consists of the wedge (WED), REC, and zinc finger (ZF) domains. In contrast, the CTD comprises the RuvC domain and a ZF domain referred to as the target nucleic-acid-binding (TNB) domain. The Cas12f dimer adopts a lobed structure, with a REC lobe and a nuclease (NUC) lobe, and the guide RNA–target DNA complex resides within the channel between them. The REC lobe is formed by the WED.1/ZF.1/REC.1 and WED.2/ZF.2/REC.2 domains of Cas12f.1 and Cas12f.2, respectively, whereas the NUC lobe consists of the RuvC.1/TNB.1 and RuvC.2/TNB.2 domains of Cas12f.1 and Cas12f.2. 

Despite its smaller size, Cas12f encompasses all the typical domains found in other Cas12 proteins [41]. For instance, it possesses several functional domains, including RuvC, WED, and REC1. Among these domains, the RuvC domain plays a crucial role of excision. In Cas12f, substrate recognition activates only one RuvC domain in the dimer, capturing the substrate within the structure. This unique feature distinguishes Cas12f from other CRISPR nucleases. [45]. Upon substrate recognition, the RuvC domain, responsible for cleaving the target DNA, undergoes a close-to-open transition in the lid motif. The WED domain is composed of a seven-stranded β-barrel flanked by an α helix and a β hairpin. It adopts an oligonucleotide/oligosaccharide-binding fold similar to that of other nucleic acid-binding proteins [46]. Though the exact function of the WED domain in Cas12f remains unclear, it is believed to participate in the recognition and binding of target DNA. Additionally, the WED domain may play a role in the conformational changes that arise during substrate recognition and cleavage by the RuvC domain. A comparison of Cas14 with the compact Cas12 protein suggested that the closest relative to Cas12f is Cas12g, which consists of 767 amino acids and belongs to branch 3 of type-V nucleases, as determined by a phylogenetic analysis [41].

The Cas12f monomer is at the NTD, containing the REC1 and WED domains, and the C-terminal holds the RuvC, REC2 (included as part of the RuvC domain in [19]), and Nuc domains (TNB domain in [19]). The primary difference between the two is the REC1 domain, which can be broken down into two subdomains: REC1N (referred to as a zinc finger or ZF domain in [19]), which features a zinc finger with a zinc ion coordinated by four cysteines (C475, C478, C500, and C503), and REC1C, a three-helix bundle that acts as the main dimerization interface of Cas12f.

The majority of Cas14-mined proteins, except for AsCas12f1, UniCas12f1, and SpCas12f, did not exhibit any DNA excision activity. These three proteins have been the focus of studies utilizing gRNA engineering to enhance their cutting efficiency in mammalian gene editing [34]. In a recent study, a comprehensive comparison was conducted to evaluate the efficiency and safety of Cas12f proteins, including AsCas12f1, CasMini, CasMINI_ge4.1, and Uni1Cas12f1_ge4.1, in comparison to commonly used Cas9, LbCas12a, and AsCas12a. The results showed that Cas12f nucleases demonstrated robust cleavage at the majority of tested sites, with deletional fragments being the predominant outcome. In contrast, Cas9 and Cas12a exhibited comparatively higher editing efficacy across most tested sites. Importantly, cells edited with Cas12f nucleases showed minimal off-target hotspots, while cells edited with Cas9 and Cas12a exhibited observable hotspots. Additionally, Cas12f nucleases reduced the occurrence of chromosomal translocations, large deletions, and integrated vectors by 2-3 fold, as compared to Cas9 and Cas12a [44]. Moreover, recently there have been advancements in utilizing the smaller-sized Cas12f to develop miniature cytosine base editors (miniCBEs) and adenine base editors (miniABEs). These newly designed miniCBEs and miniABEs have demonstrated their efficiency in correcting pathogenic mutations in cell lines. Additionally, through the delivery of an adeno-associated virus, they have successfully introduced genetic mutations in the brain in vivo. These findings highlight the potential of engineered miniCBEs and miniABEs as effective tools in gene editing for various applications. However, further research is necessary to determine the optimal conditions for their use and to thoroughly evaluate their safety and efficacy in vivo [46].

This study explores Cas14-homology proteins in prokaryotes and classifies them into three groups: Cas14-A, Cas14-B, and Cas14-U, following phylogenetic analysis. The sequence comparisons reveal high similarity at the CTD among these groups, suggesting that they may have similar modes of action. In contrast, the overall sequence similarity among all mined Cas14 sequences is only 18%. Additionally, the analysis of repeats in the CRISPR arrays of the mined Cas14 proteins reveals a wide range of repeat numbers and lengths. These results enhance our understanding of the diversity and characteristics of Cas14 proteins. The successful classification of the mined Cas14-homology proteins into three distinct clades (Cas14-A, Cas14-B, and Cas14-U) offers valuable insights for future studies and the potential application of these proteins. The identification of high similarity at the CTD end among the three clades potentially leads to more efficient and targeted gene editing. However, the low overall sequence similarity among all mined Cas14 sequences underscores the necessity for further studies to comprehensively understand the molecular basis of the differences among these proteins. This information can yield valuable insights into the specificity and efficiency of Cas14-mediated gene editing and holds implications for the evolution and adaptation of the CRISPR-Cas system in various organisms, potentially being utilized to develop novel gene-editing strategies.

## 5. Conclusions

The current study identified 93 Cas14-homology proteins within prokaryotic genomes. The mined Cas14-homology proteins were classified into three clades, namely Cas14-A, Cas14-B, and Cas14-U, based on a phylogenetic analysis. Although the key domains among the Cas14 clades were found to be conserved, specifically at the CTD, their overall sequence identities are low. Moreover, repeats associated with the mined Cas14-homology proteins were also analysed, and based on the classification of the repeats, a total of 20 putatively active proteins were found.

This study offers significant insights into the diversity and characteristics of Cas14-homology proteins and holds important implications for understanding the classification and mining of Cas14-homology proteins in proteins. Further research is necessary to comprehensively understand the molecular basis of the differences among these proteins and develop more effective gene-editing tools.

## Figures and Tables

**Figure 1 genes-14-01911-f001:**
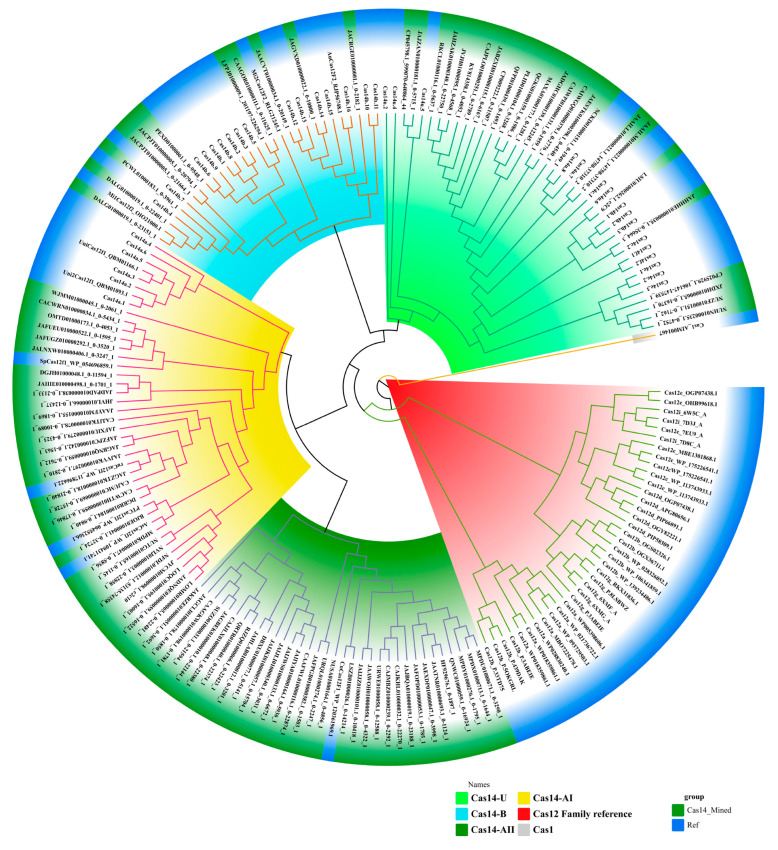
The phylogenetic tree of Cas14-homology proteins. The phylogenetic tree of mined Cas14 proteins with reference sequences from the Cas12a family and other RuvC-containing proteins. The Cas14-homology proteins were classified into three main branches (Cas14-AI, Cas14-AII, Cas14-B, and Cas14-U), as indicated with blue, purple, and orange ligands.

**Figure 2 genes-14-01911-f002:**
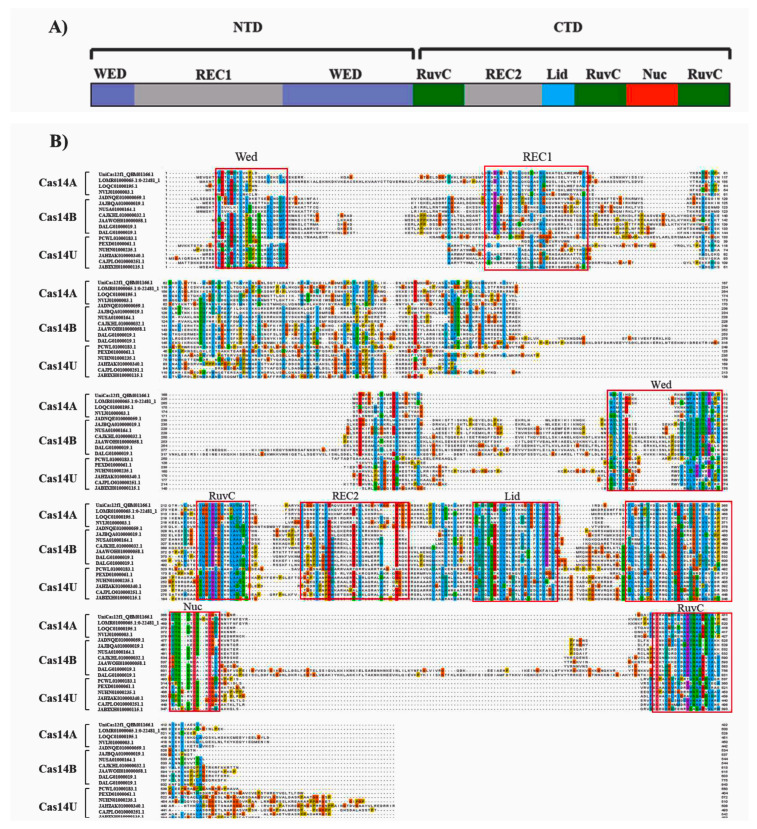
The alignments of Cas14-A, Cas14-B, and Cas14-U protein sequences. The alignment was performed with Mafft and drawn by Jalview Version 2. (**A**) Schematic of Cas14. The key domains include WED (purple), REC domain (gray), RuvC I, II, III (green), Lid (blue), and Nuc (orange). (**B**) The alignment of several sequences selected from Cas14-A, Cas14-B, and Cas14-U protein subgroups were identified based on the phylogenetic tree.

**Figure 3 genes-14-01911-f003:**
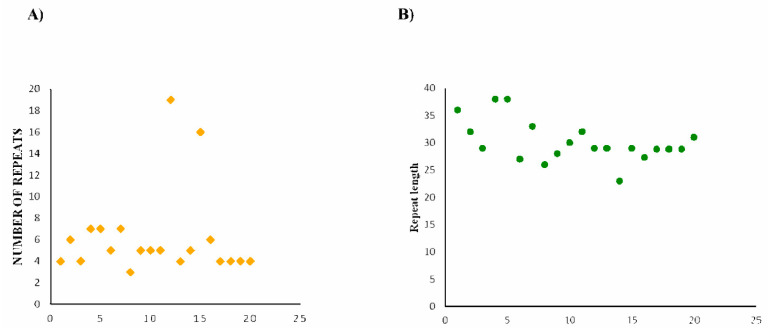
Distribution of the number of repeats and repeat length in the nucleotide sequences of mined Cas14 proteins with arrays. (**A**) The number of repeats in each nucleotide sequence. (**B**) The length (in bp) of repeats for each of the nucleotide sequences in which the array was detected (refer to Appendix A).

**Figure 4 genes-14-01911-f004:**
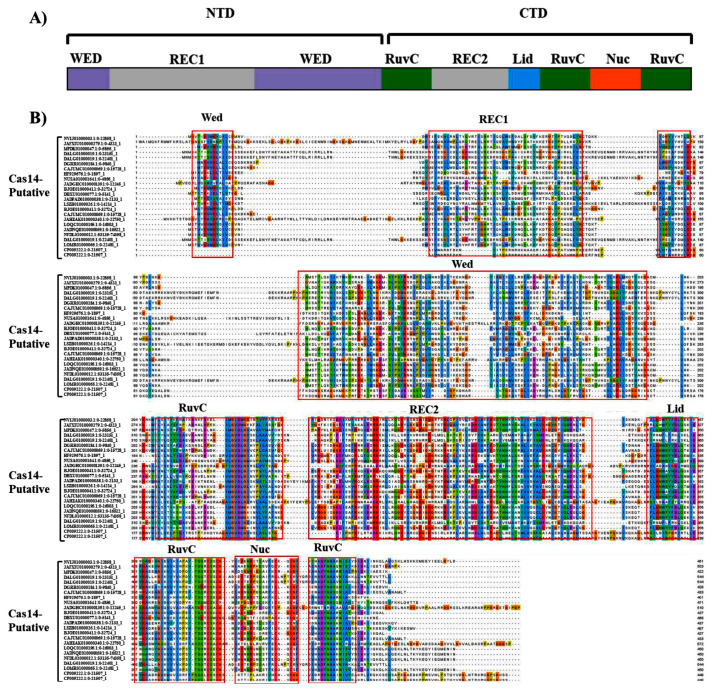
The alignments of putative active Cas14-homology protein sequences. The alignment was performed with MAFFT and drawn by Jalview Version 2. (**A**) Schematic of Cas14 (UniCas12f1). The key domains include the WED (grey), Rec1(purple), RuvC I, II, III (green), lid (blue), and Nuc (orange). (**B**) The alignment of Cas14 putative sequences with UniCas12f1. The WED, REC1, REC2, NUC, LID, and RuvC I, II, and III domains are marked using a red box.

**Table 1 genes-14-01911-t001:** Predicted repeats from the mined Cas14 sequences.

Sequence ID	Consensus Repeat	Repeat Subtype
BJOE01000041.1:0-32724_1	CTCCAAACAGAATCATGCTTCTATGACTGTTCCGAG	V-F1
CAJUMC010000069.1:0-15728_7	CTTACACCATATACCTACGCATAGTTCGAGTC	V-F1
CP009222.1:0-21507_8	GTTCTTCCCACGCACACGAAGAAGATCCC	V-F2
DALG01000019.1:0-22401_10	AGTTGCATCTCTCATCTCGTTAATTCGTGCGCTGAAAC	V-F1
DALG01000019.1:0-23151_11	AGTTGCATCTCTCATCTCGTTAATTCGTGCGCTGAAAC	V-F1
DBXU01000077.1:0-5141_12	GCTGTGACTCATAGCAAAAAAGAAGGT	V-F1
DGRR01000184.1:0-9840_14	GATTATATCTGCTTGTATGGGTATACTGCGAGA	V-F1
HF929676.1:0-1897_15	TACACACTACATAGTCATTATATAAC	V-F1
JADGHC010000139.1:0-12245_16	GGGACTTCCCCGAGCGCGAGGACGACGG	V-F2
JADPAD010000038.1:0-2133_17	GTTTAAGAATAACAATAGTTGTATTTAAAT	V-F1
JAFXIU010000279.1:0-4323_19	GTTGCAACACGCGCATAAGGATGACTTGAAGG	V-F1
MPDK01000047.1:0-5856_23	GTTCACACTCCACAAGCTAGCTCGCAAAC	V-F1
NUSA01000164.1:0-4856_24	GTTTTGAATAAACTATGTAGAATGTGAAT	V-F1
NVIJ01000003.1:0-22858_25	ATTTAAATACATCTTATGTTAGT	V-F1
LSZB01000026.1:0-14214_33	ATTTACATTTCACATAGTTAAACTAAAAC	V-F1
JAHZAK010000340.1:0-22750-1_36	GTGTTCCCCGTATGTGCGGGGGTGAGC	V-F2
LOQC01000195.1:0-16003-1_48	ATTTCAATACATCTATTGTTATGTTTTAAC	V-F1
JADNQE010000059.1:0-16522-1_52	ATTTCAATACATCTATTGTTATGTTTTAAC	V-F1
LOMR01000065.1:0-22481-1_53	ATTTCAATACATCTATTGTTATGTTTTAAC	V-F1
NFDL01000012.1:53135-74358-1_58	AGGAAAAACATAATAATAGATGTATTGAAAT	V-F1
BJOE01000041.1:0-32724_1	CTCCAAACAGAATCATGCTTCTATGACTGTTCCGAG	V-F1
CAJUMC010000069.1:0-15728_7	CTTACACCATATACCTACGCATAGTTCGAGTC	V-F1
CP009222.1:0-21507_8	GTTCTTCCCACGCACACGAAGAAGATCCC	V-F2
DALG01000019.1:0-22401_10	AGTTGCATCTCTCATCTCGTTAATTCGTGCGCTGAAAC	V-F1

**Table 2 genes-14-01911-t002:** Distribution of the number of Cas14 variants and putative Cas14 in bacterial and archaeal groups.

Group			All	Putative
Bacteria	Nitrospirae	Nitrospirae	0	0
	FCB group	Fibrobacteres	0	0
		Bacteroidetes	14	1
		Chlorobi	0	0
		Gemmatimonadetes	0	0
	PVC group	Verrucomicrobia	0	0
		Planctomycetes	0	0
		Chlamydiae	0	0
	Terrabacteria group	Deinococcus-Thermus	0	0
		Firmicutes	37	16
		Armatimonadetes	0	0
		Chloroflexi	0	0
		Actinobacteria	16	3
		Candidatus Melainabacteria	0	0
		Cyanobacteria	1	1
		Candidatus Eremiobacteraeota	0	0
	Proteobacteria	Gammaproteobacteria	0	0
		Alphaproteobacteria	2	0
		Betaproteobacteria	1	0
		unclassified Proteobacteria	0	0
		environmental samples	0	0
		delta/epsilon subdivisions	0	0
		Zetaproteobacteria	0	0
		Oligoflexia	0	0
		Acidithiobacillia	0	0
		Candidatus Lambdaproteobacteria	0	0
		Candidatus Muproteobacteria	0	0
		Hydrogenophilalia	0	0
	Aquificae	Aquificae	0	0
	Thermotogae	Thermotogae	0	0
	Deferribacteres	Deferribacteres	0	0
	Chrysiogenetes	Chrysiogenetes	0	0
	Thermodesulfobacteria	Thermodesulfobacteria	0	0
	Spirochaetes	Spirochaetes	0	0
	Fusobacteria	Fusobacteria	0	0
	Acidobacteria	Acidobacteria	0	0
	Dictyoglomi	Dictyoglomi	0	0
	Calditrichaeota	Calditrichaeota	0	0
	Nitrospinae/Tectomicrobia group	Nitrospinae/Tectomicrobia group	0	0
	Krumholzibacteriota	Krumholzibacteriota	0	0
	Caldiserica/Crysericota group	Caldiserica/Crysericota group	0	0
	Coprothermobacteria	Coprothermobacteria	0	0
	Elusimicrobia	Elusimicrobia	0	0
	Synergistetes	Synergistetes	0	0
	unclassified Bacteria	unclassified Bacteria	0	0
	environmental samples	environmental samples	0	0
Archaea	Asgard group		0	0
	Candidatus Thermoplasmatota		0	0
	DPANN group		9	3
	Euyarchaeota		1	0
	TACK group		0	0

## Data Availability

Not applicable.

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
