# Peer review of "Comparative Analysis and Phylogenetic Insights of Cas14-Homology Proteins in Bacteria and Archaea"

_genes, 2023, doi:10.3390/genes14101911_

Round 1

Reviewer 1 Report

The manuscript provide relevant characterisation of Cas14-homology proteins. Minor comments:

If possible rephrase the title to be more focused on major findings rather then techniques.

Please abbreviate the Abstract to follow the instructions of the journal.

Figure 3 could be placed to supplementary data.

Figure 5 could be shown as a Table.

L51 E.coli italics (and further)

Author Response

Thank you for the valuable comments. The comments were helpful in improving the Quality of the manuscript. 

Reviewer 2 Report

This manuscript primarily focuses on analyzing Cas14-homology and their domain disparities. While the article is well-written, there is room for improvement in terms of readability. Additionally, the figures need to be updated. Therefore, major revision is suggested. 

1. The text in Figure 1 is not clear. Please consider replacing the figure with a higher resolution image.

2. Similarly, the text in Figure 2, Figure 3, and Figure 6 is not clear. Please consider replacing these figures with higher resolution images.

3. I believe there is a typo in line 181, where it mentions the "orphan" CRISPR. If this is not a typo, could the authors provide an explanation of the meaning of "the orphan CRISPR"?

4. In lines 184-189, the authors stated that putative proteins share a major domain present in UniCas12f1. Does this mean that these putative proteins can excise target DNA like UniCas12f1? I suggest the authors add more information on the relationships between the proteins' functions and their domains. For example, clarify which domain's presence leads to which function. This would be helpful for readers who are not familiar with the field.

5. Some abbreviations, such as NTD and REC, are described with both their abbreviations and full names throughout the manuscript. To maintain consistency, I would suggest using the abbreviations once they have been introduced at the beginning of the manuscript.

6. The manuscript primarily focuses on the investigation of Ca14-homology. I wonder if the authors could also add discussion about the relationship between their sequence and excision behavior. I believe it would be more helpful for readers. 

Author Response

Thank you for taking the time and review our manuscript. Your comments were helpful in improving the quality of our manuscript.   

Round 2

Reviewer 2 Report

The authors have addressed each comments and revised their manuscript. Accept is suggested.